# Subversion of Host Cell Mitochondria by RSV to Favor Virus Production is Dependent on Inhibition of Mitochondrial Complex I and ROS Generation

**DOI:** 10.3390/cells8111417

**Published:** 2019-11-11

**Authors:** MengJie Hu, Marie A. Bogoyevitch, David A. Jans

**Affiliations:** 1Department of Biochemistry and Molecular Biology, Monash University, Monash, Victoria 3800, Australia; mengjie.hu@monash.edu; 2Department of Biochemistry and Molecular Biology, University of Melbourne, Melbourne, Victoria 3010, Australia; marieb@unimelb.edu.au

**Keywords:** respiratory syncytial virus (RSV), mitochondrial complex I, mitochondrial ROS (mtROS)

## Abstract

Respiratory syncytial virus (RSV) is a key cause of severe respiratory infection in infants, immunosuppressed adults, and the elderly worldwide, but there is no licensed vaccine or effective, widely-available antiviral therapeutic. We recently reported staged redistribution of host cell mitochondria in RSV infected cells, which results in compromised respiratory activities and increased reactive oxygen species (ROS) generation. Here, bioenergetic measurements, mitochondrial redox-sensitive dye, and high-resolution quantitative imaging were performed, revealing for the first time that mitochondrial complex I is key to this effect on the host cell, whereby mitochondrial complex I subunit knock-out (KO) cells, with markedly decreased mitochondrial respiration, show elevated levels of RSV infectious virus production compared to wild-type cells or KO cells with re-expressed complex I subunits. This effect correlates strongly with elevated ROS generation in the KO cells compared to wild-type cells or retrovirus-rescued KO cells re-expressing complex I subunits. Strikingly, blocking mitochondrial ROS levels using the mitochondrial ROS scavenger, mitoquinone mesylate (MitoQ), inhibits RSV virus production, even in the KO cells. The results highlight RSV’s unique ability to usurp host cell mitochondrial ROS to facilitate viral infection and reinforce the idea of MitoQ as a potential therapeutic for RSV.

## 1. Introduction

Respiratory syncytial virus (RSV) is the leading cause of acute lower respiratory tract infections (ALRTIs) in humans, resulting in ~33 million new cases of lower respiratory tract illness and around 55,000–199,000 deaths worldwide each year [1,2,3]. Although almost everyone has been infected before the age of two [1,2], pre-term infants, young children, immunosuppressed adults, and the elderly are all susceptible to severe infection, characterized by bronchiolitis associated with epithelial necrosis, sloughing of the airway epithelium, edema, and increased secretion of mucus [4,5,6]. RSV infection is also known to exacerbate existing pulmonary conditions, such as chronic obstructive pulmonary disease (COPD) and asthma. To date, the only approved prophylactic and therapeutic options for RSV are palivizumab (Synagis^®^) and ribavirin, respectively [7,8], both of which are restricted for use only in selected high-risk patients of RSV infection, with accompanying concerns of cost-effectiveness [8,9,10,11].

As an enveloped, non-segmented, negative-sense single-stranded RNA ((–)ssRNA) virus) of the *Pneumoviridae* family in the order of *Mononegavirales* [12,13], RSV replicates and propagates readily in the cytoplasm of infected cells. Mononegaviruses have been reported to modulate host cell mitochondrial function to facilitate viral survival, replication, and production [14,15,16,17,18]. We recently delineated RSV-induced microtubule/dynein-dependent mitochondrial perinuclear clustering and translocation towards the microtubule-organizing center in infected cells, concomitant with impaired mitochondrial respiration, loss of mitochondrial membrane potential, and increased production of mitochondrial reactive oxygen species (mtROS) [19,20]. Strikingly, agents that target microtubule integrity or the dynein motor protein or inhibit mtROS production strongly suppress RSV virus production, including in a mouse model with concomitantly reduced virus-induced lung inflammation [19]. However, the mitochondrial components targeted by RSV in this context remain unexplored.

In the present study, we employed knock-out (KO) cell lines lacking mitochondrial complex I activity [21] to examine this for the first time. The KO lines showed decreased mitochondrial respiration and enhanced mtROS and concomitantly elevated levels of wild-type (WT) RSV replication and infectious virus production. KO lines re-expressing mitochondrial complex I activity did not show this. Strikingly, blocking mtROS generation using the specific scavenger, mitoquinone mesylate (MitoQ), in the WT and KO lines resulted in inhibited RSV virus production. Together, the results highlight RSV’s unique ability to usurp host cell mtROS to facilitate viral infection and reinforce the utility of MitoQ [19] as a potential therapeutic for RSV.

## 2. Materials and Methods

### 2.1. Cell Culture, RSV Infection, and RSV Growth

Cell lines were confirmed mycoplasma-free by regular testing. They were maintained in a humidified atmosphere (5% CO_2_, 37 °C) and passaged (3-day intervals) by dissociation with trypsin-EDTA (Gibco/Thermo Fisher Scientific, Waltham, MA, USA). Vero (African green monkey kidney epithelial cells, ATCC: CCL-81, American Type Culture Collection (ATCC), Manassas, VA, USA) and human embryonic kidney (HEK) 293T cells, including WT HEK293T (ATCC: CRL-1573), CRISPR-knock-out lines of complex I α subcomplex subunit 10 (FA10), complex I β subcomplex subunit 10 (FB10), complex I β subcomplex subunit 4 (FB4), or transmembrane protein 261 (TMEM261, also known as distal membrane-arm assembly complex protein 1 (DMAC1)), as well as retrovirus-rescued lines with cDNA expression for the respective gene [21], were grown in Dulbecco’s modified Eagle’s medium (DMEM, Gibco), containing 10% heat-inactivated fetal calf serum (FCS; DKSH Australia Pty Ltd. Melbourne, Victoria, Australia), 100 U/mL penicillin (Gibco), and streptomycin (Gibco).

As in previous experiments [22], virus stocks were grown in Vero cells. HEK293T cells were grown for 12 h before infection with RSV A2 (denoted as RSV throughout) in 2% FCS/DMEM medium (multiplicity of infection (MOI) of 0.3 or 1). After 2 h, cells were washed and media replaced; cells at various times post infection (p.i.) were retained for analysis of the cell-associated infectious virus (plaque forming units) and/or viral genomes (by quantitative PCR) as per [19,22].

### 2.2. Assessment of Mitochondrial Bioenergetics and Function

The oxygen consumption rate (OCR) and extracellular acidification rate (ECAR) were monitored using the Seahorse XF96 Extracellular Flux Analyzer (Seahorse Biosciences/Agilent Technologies, Billerica, MA, USA) [23]. HEK293T cells were plated (3.5 × 10^4^ cells/well, 10% FCS/DMEM) with or without RSV infection (MOI 1, 2% FCS/DMEM, 2 h). Before the measurement, cells were washed twice with pre-warmed Seahorse assay buffer (unbuffered DMEM supplemented with 25 mM glucose, 2 mM L-glutamine, and 1 mM sodium pyruvate, pH 7.4, Seahorse Biosciences/Agilent Technologies, Billerica, MA, USA) and then equilibrated in Seahorse assay buffer (37 °C, 1 h). Respiratory parameters for basal, ATP-linked, maximal uncoupled, spare, and non-mitochondrial respiration were calculated from OCR in response to the sequential addition of 1 μM oligomycin (ATP synthase inhibitor, Seahorse Biosciences/Agilent Technologies, Billerica, MA, USA), 1 μM FCCP (carbonyl cyanide p-trifluoromethoxyphenylhydrazone, proton ionophore, Seahorse Biosciences/Agilent Technologies, Billerica, MA, USA), and a combination of 1 μM antimycin A (complex III inhibitor, Seahorse Biosciences/Agilent Technologies, Billerica, MA, USA) and 1 μM rotenone (complex I inhibitor, Seahorse Biosciences/Agilent Technologies, Billerica, MA, USA), respectively [19,23].

### 2.3. Measurement of mtROS

mtROS was detected using the mitochondria-targeted ROS sensor, flavin-rhodamine redox sensor 2 (FRR2) [19]. HEK293T cells were mock- or RSV-infected (MOI 1) or treated with complex I inhibitor rotenone (0.5 μM, 30 min), mitochondrial ROS scavenger mitoquinone mesylate (MitoQ) (1 μM, 2 h), or the vehicle control dimethyl sulfoxide (DMSO, Sigma-Aldrich, St. Louis, MO, US) [19]. Mitotracker Deep Red and FRR2 (2 μM, 15 min) with Hoechst (5 µg/mL) were added in the last 5 min before live cell imaging at 8 or 18 h using a Leica TCS SP5 channel confocal microscope with resonant scanning. The ratiometric output of FRR2 [24] (I _(Ex514)/_I _(Ex488)_, the ratio of the intensity of red emission [denoted as I] at 580 ± 20 nm upon excitation [Ex] at 514 nm *versus* 488 nm), serves as a marker for mtROS accumulation. Ratiometric I _(Ex514)/_I _(Ex488)_ images were generated by pixel-wise division of the 514 nm and 488 nm emission image channels using Fiji (https://fiji.sc/) [19]. For all samples, images were set to 32-bit float precision with a display range of min = 0.0 and max = 15.0 to facilitate comparison). To quantify the mitochondrial-localized ratio, a CellProfiler pipeline (http://cellprofiler.org/) was used, whereby a pixel-wise image of I _(Ex514)/_I _(Ex488)_ was derived by pixel-wise division of the emission image channels acquired at 514 nm and 488 nm excitation and stored as a 32-bit float image [19]. Regions containing mitochondria were then segmented from the MitoTracker Deep Red channel by applying a 5 pixel Gaussian blur and an Otsu auto-threshold [25] and then filtered to exclude all regions smaller than 1000 pixels. Segmented regions were then used to determine the mean ratiometric pixel value using the I _(Ex514)/_I _(Ex488)_ image above [19].

### 2.4. Cell Viability Assay

The LDH Release Assay cytotoxicity detection kit from Roche Applied Science (Penzberg, Bavaria, Germany) was used to quantitatively assess cell death on the basis of the amount of lactate dehydrogenase (LDH) released into the medium upon plasma membrane damage. The LDH assay was carried out as was done so previously [19,23], according to the manufacturer’s instructions.

### 2.5. Statistical Analysis

All quantitative data in this study represent the mean value ± SEM for *n* ≥ 3 (number of experiments). Significance levels were determined by one-way analysis of variance (ANOVA) unless explicitly stated otherwise (GraphPad Prism 6).

## 3. Results

### 3.1. Cells Deficient in Mitochondrial Complex I β Subcomplex Subunit 10 Show Reduced Mitochondrial Respiration and Increased RSV Virus Production; A correlation with mtROS Production

We set out to deepen our understanding of the targets of RSV infection that are involved RSV’s profound effects on the mitochondria of infected cells [19]. We decided to focus on mitochondrial respiratory complex I by examining KO lines of human embryonic kidney (HEK) 293T cells lacking the specific complex I proteins NDUFB4 or NDUFB10 (FB4^−/−^ and FB10^−/−^ cells, respectively) [21]; HEK293T cells are an accepted model of RSV infection [26,27,28]. We firstly evaluated the mitochondrial respiratory capacity of these lines using the Seahorse XF96 Extracellular Flux Analyzer as previously used [19,23]. The oxygen consumption rate (OCR), as an indicator of mitochondrial respiration, was dramatically decreased in both knock-out lines compared to the wild-type, with FB10^−/−^ cells showing a more severe defect than FB4^−/−^ cells (Figure 1A). Successive OCR measurements in the presence of oligomycin (ATP synthase inhibitor), FCCP (proton ionophore), antimycin A (mitochondrial complex III inhibitor), and rotenone (mitochondrial complex I inhibitor) (Figure 1A) enabled the key parameters of mitochondrial metabolic activity to be determined (see Figure 1B), revealing significant (*p* < 0.001) decreases (60–80%) in basal and ATP-linked OCR in both KO cells compared to the wild-type cells, with the FB10^−/−^ line showing more severe effects (Figure 1B). Clearly, cells lacking mitochondrial complex I β subcomplex subunit 10 are markedly affected in mitochondrial respiration.

To test the effect of complex I β subcomplex subunit 10 KO on RSV infection, wild-type cells, as well as the FB4^−/−^ and FB10^−/−^ lines, were infected with RSV (MOI 0.3 or 1) and assessed for infectious virus production at 24 h p.i. Strikingly, although FB4^−/−^ cells showed no difference to wild type cells in terms of infectious virus production, significantly (*p* < 0.001) higher (>3.5-fold) numbers of infectious virus were evident for the FB10^−/−^ cells (Figure 1C). Clearly, RSV infection proceeds much more robustly in cells deficient in mitochondrial complex I β subcomplex subunit 10 that have severely impaired mitochondrial respiration.

To shed light on the molecular basis of this effect, we firstly assessed the mitochondrial redox state in the respective cell lines. As previously described [19], cells were stained with the reversible mtROS sensor FRR2 [24] in parallel with Mitotracker Deep Red to visualize mitochondrial localization (Figure 2A). The fact that the oxidized form of FRR2 emits at 580 nm much more strongly upon excitation at 514 nm than at 488 nm enables ratiometric live imaging of mtROS production in situ [19,24]. Wild-type cells were treated with rotenone (complex I inhibitor) as a positive control, resulting in strong FRR2 emission in regions colocalizing with Mitotracker Deep Red (Figure 2A; 2nd row of panels); to confirm that mitochondria were responsible for this staining, we used the mtROS scavenger mitoquinone mesylate (MitoQ) [19,29,30], which strongly suppresses the actions of rotenone (Figure 2A, 3rd row of panels). Whereas the FB4^−/−^ line (4th row of panels) exhibited comparable levels of fluorescence to wild-type cells, FB10^−/−^ cells showed high levels of FRR2 fluorescence (last row of panels), consistent with elevated mtROS production.

Quantitative analysis of the ratiometric images confirmed significantly (*p* < 0.001) elevated (>70%) levels of mtROS in FB10^−/−^ cells compared to wild-type cells (Figure 2B). FB4^−/−^ cells, in contrast, showed results comparable to those for wild-type cells (Figure 2B); thus, cells knocked out for NDUFB10 that show elevated RSV virus production compared to wild type (Figure 1C) have enhanced mtROS, in contrast to FB4^−/−^ cells, which have essentially normal mtROS levels; presumably this relates to the fact that NDUFB10 is integrally involved in Complex I formation/proton translocation [31], whereas NDUFB4′s role is more peripheral in helping mediate higher order interactions (e.g., with Complex III [32]). Significantly, the results for the KO lines reveal a correlation between elevated mtROS generation and increased RSV virus production, with the clear implication that mtROS contributes integrally to infectious RSV production.

### 3.2. Correlation of Infectious RSV Virus Production with Impaired Mitochondrial Respiration and Increased mtROS Generation in Cell Lines Knocked Out for Specific Mitochondrial Genes and/or Retrovirus-Rescued

We decided to confirm the above effects by examining complex I KO lines NDUFA10 (FA10^−/−^) and NDUFB10 (FB10^−/−^) in parallel with retrovirus-rescued derivative lines expressing the cDNAs for NDUFA10 (FA10*) or NDUFB10 (FB10*), respectively [21]. We also included an additional non-complex I control in the form of a line knocked out for non-complex I mitochondrial transmembrane protein TMEM261 (TMEM261^−/−^) and its retrovirus-rescued (TMEM261*) derivative [21]. All lines were infected with RSV (MOI 0.3 or 1) and assessed for viral replication and infectious virus production at 24 h p.i. (Figure 3A,B). Strikingly, the FA10^−/−^ and FB10^−/−^ lines showed significantly (*p* < 0.001) higher (>50%) numbers of viral genomes and infectious virus titers compared to both wild-type cells and their respective retrovirus-rescued counterparts, FA10* and FB10* (Figure 3AB). In contrast, TMEM261^−/−^ cells had a somewhat reduced number of viral genomes and virus titers compared to both wild-type and TMEM261* (Figure 3A,B).

Seahorse XF96 Extracellular Flux Analyzer measurements were then used to demonstrate firstly that the FA10^−/−^ line, similar to the FB10^−/−^ line, was strongly (~80%) impaired in mitochondrial respiratory function compared to wild-type cells, whereas the TMEM261^−/−^ line showed significantly (*p* < 0.05) increased (~20%) activity (Figure 3C). OCR measurements indicated that the retrovirus-rescued cells in all cases strongly resembled wild-type cells (Figure 3C). The results indicate an inverse relationship between respiratory activity and RSV replication and infectious virus production, with complex I subunit deletion severely impacting respiratory activity but markedly enhancing infectious virus production.

Analysis of the Mitochondrial redox states by staining using FRR2 in parallel with Mitotracker Deep Red as above (Figure 4) showed a clear correlation between mtROS generation and virus production. As above, wild-type cells treated with rotenone were used as a positive control for high mtROS levels, with strong FRR2 emission in regions colocalizing with Mitotracker Deep Red (Figure 4A; 2nd row of panels). Compared to wild-type cells (1st row of panels), as well as FA10* and FB10* cells (4th and 6th rows of panels), FA10^−/−^ (3rd row of panels), and FB10^−/−^ (5th row of panels), cells showed high levels of FRR2 fluorescence. In contrast, TMEM261^−/−^ and TEME261* cells (last two rows of panels) exhibited levels of fluorescence comparable to wild-type cells. Quantitative analysis of the ratiometric images confirmed significantly (*p* < 0.001) elevated (70–80%) levels of mtROS in FA10^−/−^ and FB10^−/−^, but not in TMEM261^−/−^ cells, compared to both wild-type and the respective retrovirus-rescued cells (Figure 4B). Clearly, cells knocked out for NDUFA10 or NDUFB10 with decreased mitochondrial respiration have enhanced mtROS production and elevated RSV virus production as a result. This is in contrast to TMEM261^−/−^ cells, which have essentially normal mitochondrial respiration and mtROS levels, and retrovirus-rescued complex I KO lines, where mitochondrial respiration and mtROS is normalized and RSV virus production is comparable to wild-type cells. Clearly, complex I activity is central to RSV infection, with reduced activity leading to impaired mitochondrial respiration, increased mtROS generation, and facilitated virus production. This underlines the importance to the infectious cycle of RSV’s impact on host mitochondria.

### 3.3. The mtROS Scavenger Mitoquinone Mesylate (MitoQ) Protects Against RSV Infection in Human Embryonic Kidney Cells as well as Human Alveolar Epithelial Cells

The data above clearly implies a causative role for mtROS in facilitating RSV infectious virus production. To test this formally, we decided to test whether blocking mtROS generation during RSV infection could limit RSV infection, as previously done, using the mtROS scavenger MitoQ [19]. Wild-type and KO cells were infected with RSV (MOI 1) and then treated with MitoQ at different times p.i., prior to cell lysis and subsequent quantitation of viral replication (qPCR) and infectious virus production (plaque assay) (Figure 5). Results were unequivocal, showing significant (*p* < 0.001) reduction of both viral replication and infectious virus production in both wild-type and KO cells in the presence of MitoQ. Treatment with MitoQ from 8 h p.i. had essentially the same effect in both lines, leading to >10- and >100-fold lower levels of RSV genomes and infectious virus titer, respectively (Figure 5A,B). MitoQ treatment from 18 h had a less profound effect, but still significantly (*p* < 0.001) reduced RSV genomes and the infectious virus titer by ~2- and almost 100-fold, respectively (Figure 5A,B).

As already indicated, HEK293T cells are an accepted infectious model for RSV [26,27,28]; our previous work with cells of the A549 human alveolar epithelial line, as well as primary human bronchiolar epithelial cells, indicated clear effects of RSV infection on mitochondrial respiratory activity and mtROS production [19], closely paralleling those observed here for HEK293T cells. We were keen to establish the relevance of the results here to other models of infection, such as A549 cells, with respect to protection by MitoQ against infection, over a longer, more physiological course of infection. We infected A549 with RSV for up to 8, 12, 24, and 48 h and added MitoQ for the last 8 h in each case (Figure 6). The results show that an 8 h addition of MitoQ is able to significantly (*p* < 0.001) suppress RSV replication (Figure 6A) and infectious virus production (Figure 6B), even at late stages of infection (i.e., 40 h p.i.); strikingly, 8 h treatment with MitoQ 40 h p.i. could still reduce infectious virus production almost 100-fold. To confirm that MitoQ’s effects in this context are not through impacting cell viability, we assessed the release of the cytosolic enzyme lactate dehydrogenase (LDH) into the culture medium, indicative of cell death (Figure A1 in Appendix B). None of the MitoQ concentrations used significantly increased LDH release compared to that of controls in either the absence or presence of RSV infection, confirming that MitoQ is not cytotoxic; this is entirely consistent with other studies [19], with toxicity also shown to be absent in animal models and humans [34,35,36,37,38]. Overall, the results underline the relevance of the results here, with respect to the key role of mitochondrial Complex I in RSV infection to the development of anti-RSV therapies, with MitoQ of great potential as a viable option to treat established RSV infection (see Discussion).

## 4. Discussion

This is the first study to use KO cells to establish the importance of mitochondrial complex I in RSV infection. Deficiency of either of the specific complex I proteins, NDUFA10 and NDUFB10, that are integral to electron transport and Complex I assembly [21,39,40] results in elevated mtROS levels, compared to wild type (Figure 4), and enhanced virus production (Figure 3B). This is in stark contrast to the knock-out of other mitochondrial proteins, such as non-complex I protein TMEM261 or complex I component NDUFB4, which has more peripheral roles in super complex formation [31,32,41]; in these cases, lack of expression does not impact mtROS levels (Figure 4), and RSV virus production is essentially normal (Figure 3B). It is significant in this context that NDUFB10 deficiency has previously been associated with increased mtROS and compromised mitochondrial respiration [31,42], while mutations in NDUFA10 have been found in mitochondrial disorders, such as Leigh disease [41,42,43]. Future studies could include studying viral growth kinetics in Complex I KO cells to elucidate the precise mechanism by which reduced mtROS favors RSV virus production.

It is known that RSV infection interferes with many key cellular functions, but many of these appear to relate, as in the present study, to host cell mitochondria. Unbiased proteomic studies in RSV-infected cells, for example, reveal a large-scale impact on the abundance of a number of mitochondrial proteins, including complex I proteins, as well as voltage-dependent anion channel proteins, mitochondrial antioxidant enzymes, and the prohibitin subunits that play roles in regulating mitochondrial morphology, distribution, and biogenesis [44,45,46]. RSV’s non-structural (NS) proteins are known to degrade signaling proteins required for interferon induction or response pathways early in infection [17]; such activity requires the formation of a NS-associated degradative complex on the mitochondria in a mitochondrial antiviral signaling adaptor (MAVS)-dependent manner [17]. It seems clear that the host cell mitochondria are a key target of RSV action, consistent with our recent report documenting that RSV infection induces a staged redistribution of mitochondria concomitant with compromised mitochondrial respiration and enhanced mtROS production in A549 human alveolar epithelial cells [19]; that these effects on mitochondria enhance viral pathogenesis [17,19] further supports the idea that host cell mitochondria are a key target of RSV action. Our results here indicate for the first time that mitochondrial Complex I is central to this and the effects on decreased mitochondrial respiration and enhanced mtROS levels in particular [19]; that reduced Complex I activity leading to decreased mitochondrial respiration and increased mtROS favors RSV infection implicates Complex I as a key target of RSV in infected cells (see model in Figure 7), making Complex I a potential therapeutic target for anti-RSV intervention strategies. Importantly in this context, the mtROS scavenger, MitoQ, can limit RSV infection, even in complex I-deficient cells (Figure 5), confirming MitoQ’s utility as a viable therapeutic to combat RSV in the future (see also [19] and below).

The study here uses HEK293T cells/KO derivatives thereof as the infectious model to establish the key role of Complex I in the impact of RSV infection on mitochondrial function. We show that RSV infection strongly impacts mitochondrial respiratory function, leading to elevated mtROS production, and that the mtROS scavenger, MitoQ, in reducing mtROS in infected cells, can lead to reduced RSV replication and virus production. This aligns perfectly with our previous study [19], using more physiological models of RSV infection, including A549 human alveolar epithelial cells and primary human bronchiolar epithelial cells, where RSV infection strongly depresses mitochondrial respiratory function and elevates mtROS levels, which can be prevented with MitoQ. We also show that MitoQ protects against RSV here, even when added 40 h post-infection (Figure 6), implying that MitoQ can be effective in reducing the viral load, even in an established RSV infection. We have previously shown that MitoQ can protect against RSV in a mouse model of RSV infection [19], and the experiments here (Figure A1) and in previous cell and mouse models [19] have shown that MitoQ does not have toxic side effects. Most importantly, MitoQ has been used safely in humans [37,38]; it is clear that, taking together the results here and from our previous work [19], MitoQ or comparable mitochondrial scavengers should be considered seriously in the development of anti-RSV therapeutic strategies in the future.

## Figures and Tables

**Figure 1 cells-08-01417-f001:**
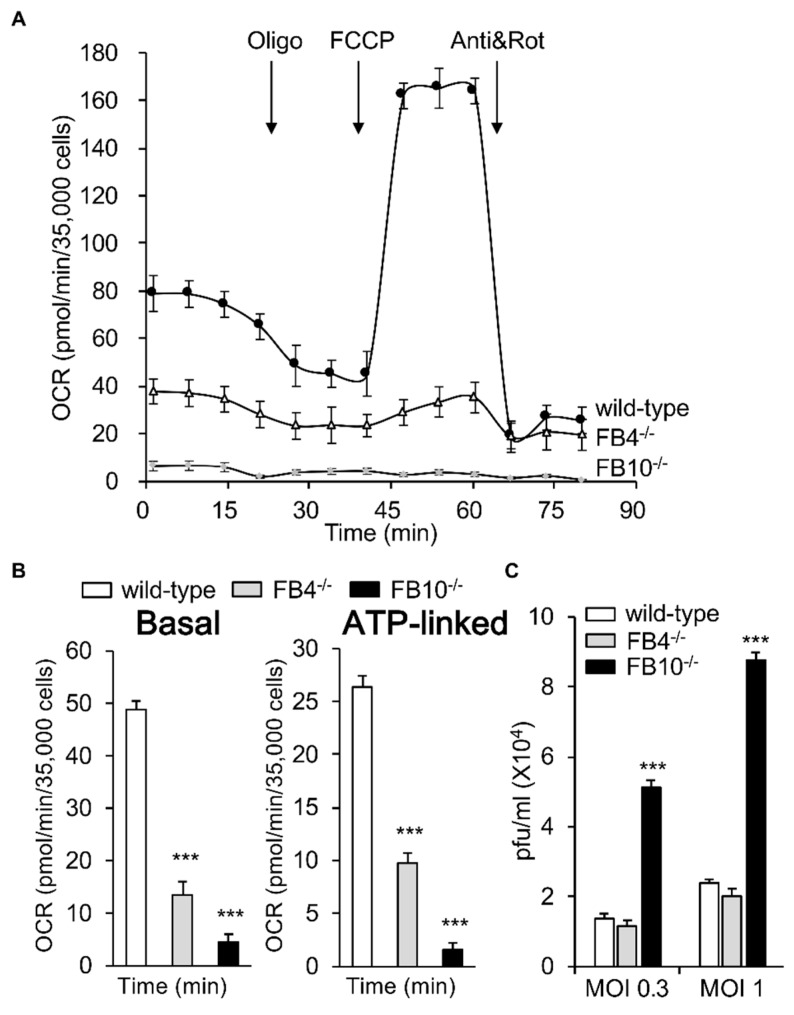
Knock-out (KO) of specific mitochondrial complex I β subcomplex subunit 10 results in reduced mitochondrial respiration; enhanced respiratory syncytial virus (RSV) virus production. Human embryonic kidney (HEK)293T cells and KO derivatives lacking Complex I β subcomplex subunit 4 (FB4^−/−^) or 10 (FB10^−/−^) [21] were (**A**,**B**) assessed for their mitochondrial bioenergetic activities using the Seahorse XF96 Extracellular Flux Analyzer. (**A**) An example of a typical oxygen consumption rate (OCR) obtained in these experiments. OCR was measured in real time upon sequential additions of ATP synthase inhibitor oligomycin (Oligo, 1 mM), proton ionophore carbonyl cyanide p-trifluoromethoxyphenylhydrazone (FCCP, 1 mM), mitochondrial complex III inhibitor antimycin A (Anti, 1 mM), and mitochondrial complex I inhibitor rotenone (Rot, 1 mM). (**B**) Mitochondrial respiratory parameters of basal and ATP-linked respiration were determined as previously [19,23]. (**C**) Cell lines were infected with RSV (multiplicity of infections (MOIs) indicated) for 24 h prior to analysis for cell-associated virus by plaque assays to determine infectious virus (plaque forming units pfu/mL). Results represent the mean ± SEM (*n* = 3 independent experiments, each performed in triplicate). ****p* < 0.001 compared to the wild-type cells.

**Figure 2 cells-08-01417-f002:**
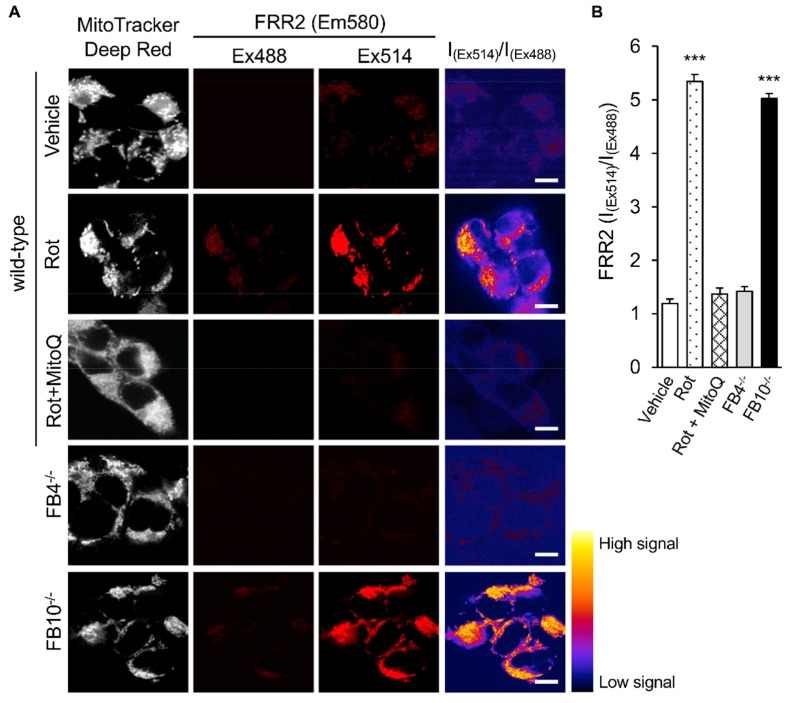
Knock-out of mitochondrial complex I β subcomplex subunit 10 results in elevated mitochondrial reactive oxygen species (mtROS) generation. HEK293T cells and KO derivative lines FB4^−/−^ or FB10^−/−^ were assessed for mtROS generation. Wild-type cells treated with mitochondrial Complex I inhibitor rotenone (Rot, 0.5 µM), Rot (0.5 µM) for 1 h, followed by mitoquinone mesylate (MitoQ) (1 µM) for a further 1 h or a dimethyl sulfoxide (DMSO) vehicle for 2 h are shown for comparison. (**A**) Cells were stained for Mitotracker Deep Red (white; 100 nM, 15 min) and the mitochondria-specific ROS probe flavin-rhodamine redox sensor 2 (FRR2, red; 2 µM, 15 min), followed by live cell imaging by resonant scanning confocal laser scanning microscopy (CLSM). Colocalisation for Mitotracker Deep Red and flavin-rhodamine redox sensor 2 (FRR2) staining at either Ex488 or Ex514 was >85% (Pearson correlation coefficient [33]) across all samples (25–30 cells/sample). The I_(Ex514)_/I_(Ex488)_ images (far right) are calculated by a pixel-wise division of the 514 nm (third column) and 488 nm (second column) emission image channels and represented in pseudo-colors (intensity color key displayed lower right). Results are typical of *n* = 3 independent experiments. In all panels, scale bar = 10 µm. (**B**) FRR2 (I_(Ex514)_/I_(Ex488)_) was calculated for the mitochondrial regions defined by Mitotracker Deep Red staining in the I_(Ex514)_/I_(Ex488)_ images, such as those in (**A)** using a custom CellProfiler pipeline (see Methods). Results represent the mean + SEM for *n* = 3 independent experiments, where each experiment analyzed 25–30 cells per sample. *** *p* < 0.001 compared to the wild-type cells.

**Figure 3 cells-08-01417-f003:**
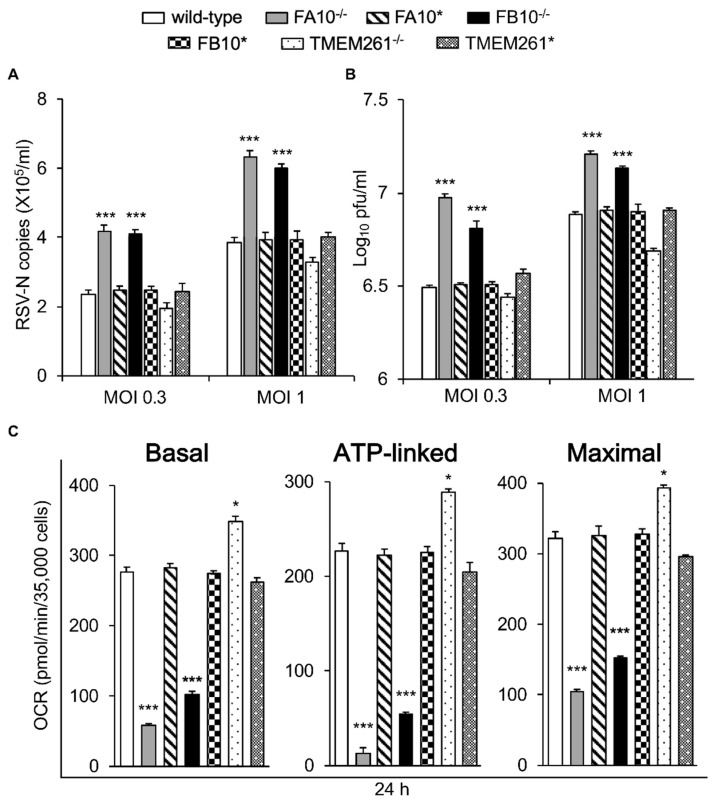
Correlation of infectious RSV virus production with impaired mitochondrial respiration in cell lines knocked out for specific mitochondrial genes and/or retrovirus-rescued. KO derivatives thereof lacking Complex I α subcomplex subunit 10 (FA10^−/−^), FB10^−/−,^ or transmembrane protein 261 (TMEM261^−/−^), as well as retrovirus-rescued controls (FA10*, FB10*, and TMEM261*) were (**A** and **B**) infected with RSV (MOI indicated) for 24 h prior to analysis for the cell-associated virus by (**A**) qPCR and (**B**) plaque assays to determine the viral RNA copy number and infectious virus (plaque forming units pfu/mL), respectively, or (**C**) assessment for their mitochondrial bioenergetic activities as per Figure 1B. Results represent the mean ± SEM (*n* = 3 independent experiments, each performed in triplicate). *** *p* < 0.001, * *p* < 0.05 compared to the rescued or wild-type cells.

**Figure 4 cells-08-01417-f004:**
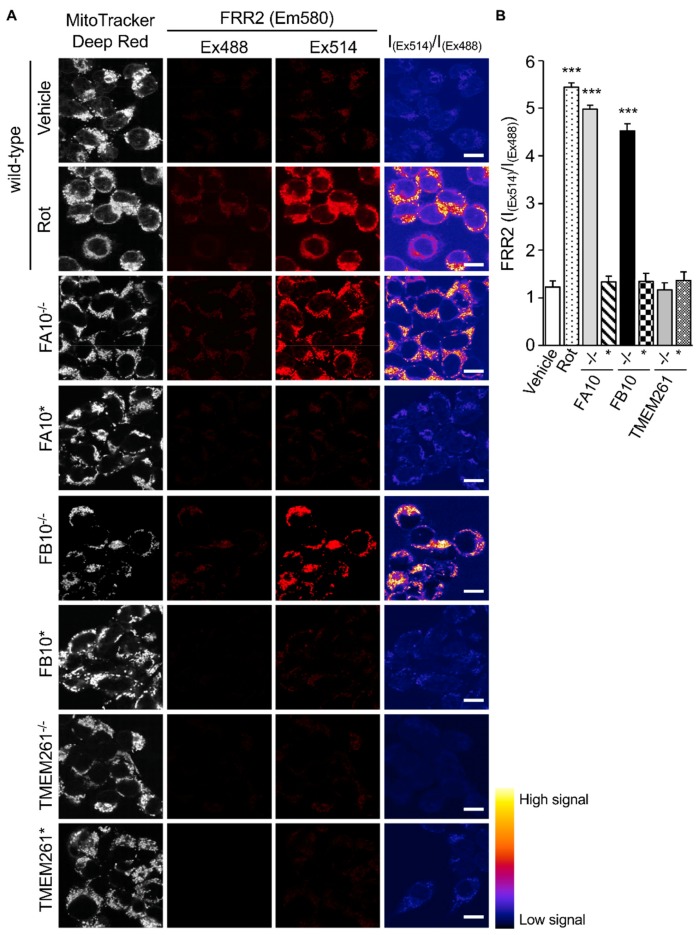
Human embryonic kidney cells lacking Complex I activity show elevated mtROS production. HEK293T cells and KO derivatives, FA10^−/−^, FB10^−/−^, or TMEM261^−/−^, as well as retrovirus-rescued controls thereof, FA10*, FB10*, or TMEM261*, were assessed for mtROS generation. (**A**) Cells were stained for Mitotracker Deep Red and the mitochondria-specific ROS probe FRR2, as per Figure 2A. The ratiometric output images of I_(Ex514)_/I_(Ex488)_ were calculated, as per Figure 2A. (**B**) FRR2 (I_(Ex514)_/I_(Ex488)_) was calculated for the mitochondrial regions defined by Mitotracker Deep Red staining in the I_(Ex514)_/I_(Ex488)_ images, such as those in **A**, using a custom CellProfiler pipeline, as per Figure 2B. Results represent the mean ± SEM for *n* = 3 independent experiments, where each experiment analyzed 25–30 cells per sample. *** *p* < 0.001 compared to the rescued or wild-type cells.

**Figure 5 cells-08-01417-f005:**
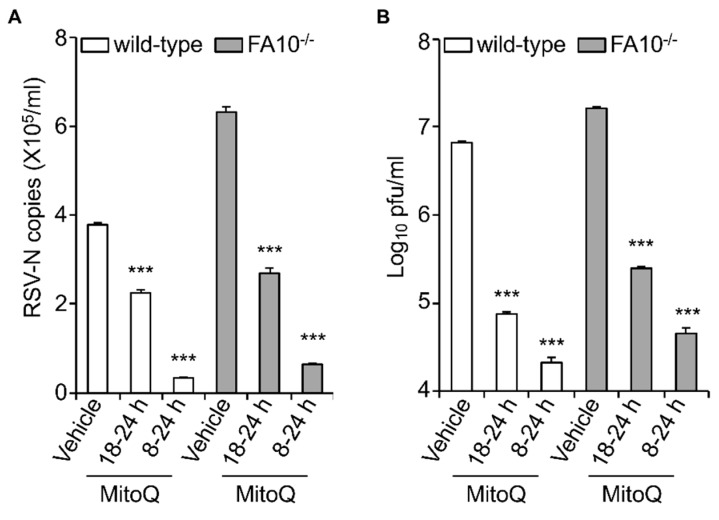
The mtROS scavenger mitoquinone mesylate (MitoQ) protects against RSV infection in human embryonic kidney cells. RSV-infected (MOI 1) wild-type or FA10^−/−^ HEK293T cells were treated with MitoQ (1 µM) for the times indicated, followed by cell lysate preparation, with (**A**) qPCR and (**B**) plaque assay performed to determine the viral RNA copy number and infectious virus (plaque forming units pfu/mL), respectively. Results shown represent the mean ± SEM (*n* = 3 independent experiments, each performed in triplicate). Two-way analysis of variance (ANOVA) analysis was performed. *** *p* < 0.001 compared to the vehicle-treated respective cell lines. RSV genome copies and virus production levels were also statistically significant between wild-type and FA1^−/−^ HEK293T cells (*** *p* < 0.001).

**Figure 6 cells-08-01417-f006:**
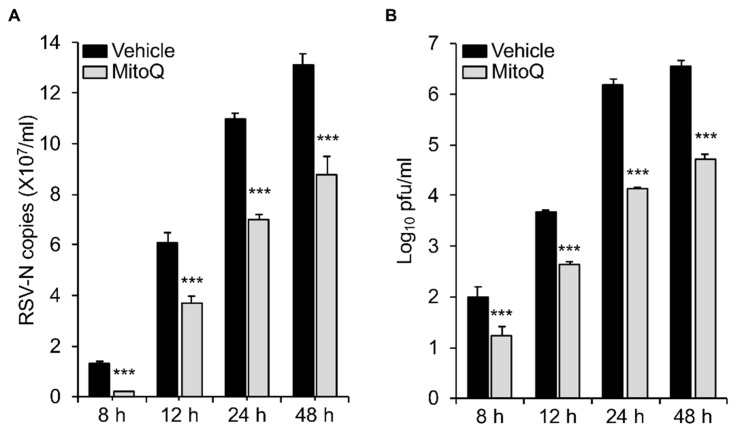
The mtROS scavenger mitoquinone mesylate (MitoQ) protects against RSV infection in A549 human adenocarcinomic alveolar basal epithelial cells. RSV-infected (MOI 1) A549 cells were treated with DMSO (vehicle) or MitoQ (1 µM) for the last 8 h before virus harvest and cell lysate preparation. (**A**) qPCR and (**B**) plaque assays were performed to determine the viral RNA copy number and infectious virus (plaque forming units pfu/mL). Results shown represent the mean ± SEM (*n* = 3 independent experiments, each performed in triplicate). *** *p* < 0.001 compared to the vehicle.

**Figure 7 cells-08-01417-f007:**
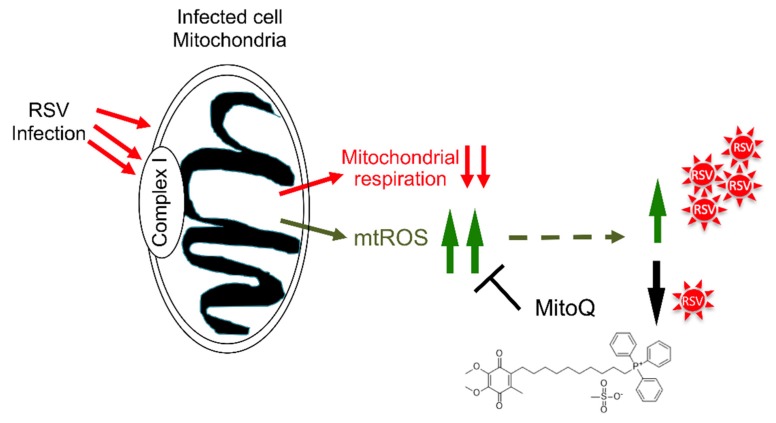
Schematic representation of the impact of RSV on host cell mitochondria. RSV infection leads to multiple impacts on the host cell mitochondria [19], in part through targeting complex I, including inhibition of host mitochondrial respiratory activity and enhanced mtROS generation to promote RSV infectious virus production. Blocking mtROS production using the mitochondrially-targeted antioxidant MitoQ limits RSV infection.

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
