# Peer review of "Subversion of Host Cell Mitochondria by RSV to Favor Virus Production is Dependent on Inhibition of Mitochondrial Complex I and ROS Generation"

_cells, 2019, doi:10.3390/cells8111417_

Round 1
Reviewer 1 Report
This manuscript reports on the importance of the relationship between mitochondrial complex 1 subunit, mitochondrial reactive oxygen species (mtROS) and RSV production at 24 hours post-inoculation. The manuscript shows conclusively using KO cells lacking mitochondrial complex 1 activity a significant decrease in oxygen consumption rate (OCR), a surrogate marker for mitochondrial activity, and KO cells that lacked the FB10 specific complex 1 protein had the greatest reduction in OCR. This reduction was associated with a significant increase (~3 to 4 fold increase) in RSV production at 24 hours post-inoculation in the FB10 KO cells compared to wild type cells. The manuscript goes on to demonstrate that FB10 KO were associated with significantly elevated levels of mtROS, and similar findings were observed when wild type cells were pretreated with rotenone, a mitochondrial complex I inhibitor. Both KO cells that lacked either FB10 or FA10 specific complex 1 protein but not TMEM261 had decreased OCR, increased RSV production and increased mtROS. Lastly post-treatment of FA10 KO cells with MitoQ (mitoquinone mesylate), a mtROS scavenger, resulted in 10 and 100-fold decrease in RSV genomes and infectious virus, respectively.
This is a well written manuscript with convincing data on the relationship between RSV infection and its role in utilizing the mitochondrial complex 1 subunit to enhance viral production early in the virus life cycle. What is not clear is if usurping the mitochondrial complex 1 activity on virus production continues beyond the 24 hour post-inoculation period. This is relevant if anti-viral therapeutics are to be developed that target mtROS. Additional data that would strengthen the manuscript would be an RSV growth kinetics at various time points such as 6, 24, 48, 72 and possibly 96 hours post-inoculation. Anti-viral drugs that have shown promise in human challenge models have failed because of inability to impact clinical illness later into the course of infection.
A second issue that needs to be address is the potential for cellular toxicity with MitoQ. Data should be presented that the cause of reduced virus production in MitoQ treated cells was not related to drug toxicity. A dose-response MitoQ experiment on cellular toxicity, mtROS quenching, and virus production would convince the readers of the relevance of mtROS to the RSV life-cycle and virion production.
Other minor comments
Abstract.
RSV is not the most common cause of respiratory infection in the populations mentioned, that honor goes to rhinovirus. Suggest something like “RSV is an an important cause of severe respiratory infection…”
Introduction.
Line 32. The number normally cited are ~33 million new cases of lower respiratory tract illness and around 55,000 to 199,000 deaths worldwide each year are attributed to RSV. Please also cite Shi Ting Lancet 2017 article.
Line 39-40. “palivizumab ….with questionable efficacy and safety profile” is absolutely an incorrect statement. Please revise. Phase III studies have clearly demonstrated the safety and efficacy of palivizumab immunoprophylaxis in preventing severe RSV infection in high risk preterm infants and those with chronic lung disease and hemodynamically significant congenital heart disease.
Materials and Methods
Line 91. Please spell out MitoQ and DMSO since it was the first time it was used and indicate the function of MitoQ, similarly as performed for the other inhibitors. That is very helpful to the readers.
Figure 5b. Both wild type and FA10 KO cells gave very similar levels of virus production by the plaque assay and N copy numbers at 24 hours. Looking at Figures 1 and 3 and 5b there appears to be some variability in the enhance virus production observed between wild type and FB10 or FA10 KO cells. This should be address in the discussion section and also stresses the need for studies on the full virus growth kinetics in relation to mitochondrial complex 1 activity.
Author Response
Dear Prof. Wang,
I am submitting a revised version of our manuscript Cells-617641R1:
“Subversion of Host Cell Mitochondria by RSV to Favour Virus Production is Dependent on Inhibition of Mitochondrial Complex I and ROS Generation”
by MengJie Hu, Marie A. Bogoyevitch and David A. Jans
for publication in Cells.
We would firstly like to thank the Reviewers and yourself for helping contribute to making our manuscript a more complete evocation of work that the Reviewers recognize as novel. We have made amendments in the manuscript to address all of the issues raised by the Reviewers, including addition of a new figure (Figure 6) as well as a supplementary figure to address questions from both reviewers, and expansion of the paper’s Discussion section. We are confident that the manuscript now satisfies the criteria for publication. The point-by-point responses to the Reviewers’ comments are below.
Reviewer #1
This manuscript reports on the importance of the relationship between mitochondrial complex 1 subunit, mitochondrial reactive oxygen species (mtROS) and RSV production at 24 hours post-inoculation. The manuscript shows conclusively using KO cells lacking mitochondrial complex 1 activity a significant decrease in oxygen consumption rate (OCR), a surrogate marker for mitochondrial activity, and KO cells that lacked the FB10 specific complex 1 protein had the greatest reduction in OCR. This reduction was associated with a significant increase (~3 to 4 fold increase) in RSV production at 24 hours post-inoculation in the FB10 KO cells compared to wild type cells. The manuscript goes on to demonstrate that FB10 KO were associated with significantly elevated levels of mtROS, and similar findings were observed when wild type cells were pretreated with rotenone, a mitochondrial complex I inhibitor. Both KO cells that lacked either FB10 or FA10 specific complex 1 protein but not TMEM261 had decreased OCR, increased RSV production and increased mtROS. Lastly post-treatment of FA10 KO cells with MitoQ (mitoquinone mesylate), a mtROS scavenger, resulted in 10 and 100-fold decrease in RSV genomes and infectious virus, respectively.
This is a well written manuscript with convincing data on the relationship between RSV infection and its role in utilizing the mitochondrial complex 1 subunit to enhance viral production early in the virus life cycle. What is not clear is if usurping the mitochondrial complex 1 activity on virus production continues beyond the 24 hour post-inoculation period. This is relevant if anti-viral therapeutics are to be developed that target mtROS. Additional data that would strengthen the manuscript would be an RSV growth kinetics at various time points such as 6, 24, 48, 72 and possibly 96 hours post-inoculation. Anti-viral drugs that have shown promise in human challenge models have failed because of inability to impact clinical illness later into the course of infection.
We thank the Reviewer for this suggestion; we now include a new figure (Figure 6), documenting the fact that mtROS scavenger MitoQ can protect against RSV infection even when added 40 h p.i. (Section 3.3 lines 282-289). The results confirm that targeting mtROS even later into the course of infection can help inhibit virus replication/production, emphasising MitoQ’s potential as a viable anti-RSV agent. We thank the Reviewer for encouraging us to perform this experiment.
A second issue that needs to be address is the potential for cellular toxicity with MitoQ. Data should be presented that the cause of reduced virus production in MitoQ treated cells was not related to drug toxicity. A dose-response MitoQ experiment on cellular toxicity, mtROS quenching, and virus production would convince the readers of the relevance of mtROS to the RSV lifecycle and virion production.
Previous studies have documented the lack of cytotoxity of MitoQ, including Phase II clinical trial, but to satisfy the Reviewer, we now include a new figure (Figure S1) documenting the minimal effects on cell viability of MitoQ in mock- and RSV-infected cells. The results confirm that the observed reduction in viral replication/production is not a result of reduced cell viability (Section 3.3 lines 289-295). We thank the Reviewer for encouraging us to present this analysis.
Minor comments:
Abstract.
RSV is not the most common cause of respiratory infection in the populations mentioned, that honor goes to rhinovirus. Suggest something like “RSV is an important cause of severe respiratory infection…”
We have revised the sentence in the abstract as requested.
Introduction - Line 32. The number normally cited are ~33 million new cases of lower respiratory tract illness and around 55,000 to 199,000 deaths worldwide each year are attributed to RSV. Please also cite Shi Ting Lancet 2017 article.
We have amended the text (Section 1 lines 32/3), and included the citation requested.
Line 39-40. “palivizumab ….with questionable efficacy and safety profile” is absolutely an incorrect statement. Please revise. Phase III studies have clearly demonstrated the safety and efficacy of palivizumab immunoprophylaxis in preventing severe RSV infection in high risk preterm infants and those with chronic lung disease and hemodynamically significant congenital heart disease.
We have amended the sentence to satisfy the Reviewer (Section 1 lines 39-41).
Materials and Methods- Line 91. Please spell out MitoQ and DMSO since it was the first time it was used and indicate the function of MitoQ, similarly as performed for the other inhibitors. That is very helpful to the readers.
We now spell out MitoQ, DMSO and functions of the inhibitors used in the study (Section 2.3 lines 91-93), as requested.
Figure 5b. Both wild type and FA10 KO cells gave very similar levels of virus production by the plaque assay and N copy numbers at 24 hours. Looking at Figures 1 and 3 and 5b there appears to be some variability in the enhance virus production observed between wild type and FB10 or FA10 KO cells. This should be address in the discussion section and also stresses the need for studies on the full virus growth kinetics in relation to mitochondrial complex 1 activity.
We have now performed two-way ANOVA analysis, and confirmed that the increase in the virus production level in FB10 KO (vehicle control) compared to that of wild-type (vehicle control) cells is statistically significantly in Figure 5A and B (Section 3.3 lines 276-278), consistent with the enhanced virus production observed between wild-type and FB10 or FA10 KO cells in Figures 1 and 3. We did not compare the virus replication/production between wild-type and FA10 KO under the vehicle-treated condition because we were focusing on the MitoQ protective effects on individual cell lines. We thank the Reviewer for pointing out this oversight. We have also stressed the need for future studies on virus growth kinetics in relation to mitochondrial complex I activity (Section 4 lines 320-321).
We thank the Reviewers again for their important contributions to making our manuscript a more rigorous and clearer evocation of work that is of interest to the broad readership of Cells.
We thank you for your assistance, and in anticipation of receiving acknowledgment of receipt of the manuscript.
Yours sincerely,
Prof. David A. Jans
PS. In addition to the revised manuscript (“Main Article File”) we have submitted a “Related Manuscript File” that highlights in red all of our changes to the original manuscript listed above. We have also submitted a Supplementary Material file which is all new data.
PPS. We have now put all of our references into the MDPI Citations Style Guide, as requested.
Reviewer 2 Report
See attached file

Author Response
Dear Prof. Wang,
I am submitting a revised version of our manuscript Cells-617641R1:
“Subversion of Host Cell Mitochondria by RSV to Favour Virus Production is Dependent on Inhibition of Mitochondrial Complex I and ROS Generation”
by MengJie Hu, Marie A. Bogoyevitch and David A. Jans
for publication in Cells.
We would firstly like to thank the Reviewers and yourself for helping contribute to making our manuscript a more complete evocation of work that the Reviewers recognize as novel. We have made amendments in the manuscript to address all of the issues raised by the Reviewers, including addition of a new figure (Figure 6) as well as a supplementary figure to address questions from both reviewers, and expansion of the paper’s Discussion section. We are confident that the manuscript now satisfies the criteria for publication. The point-by-point responses to the Reviewers’ comments are below.
Reviewer #2
In this manuscript, the authors follow up on their recent finding of RSV-induced changes in cell mitochondria localization and mitochondria-derived ROS linked to changes in virus replication by showing that mitochondrial complex I is key to this effect on the host cell, using KO cells and reconstituted cells for component of that complex.
Major comments:
Though HEK293 cells are often used for manipulation of cellular signaling, they are not a relevant cell line when it comes to RSV infection. The authors have been using A549 cells and an immortalized airway epithelial cell line for their previous studies and should confirm some of their findings in one of these cell lines.
HEK293T cells are accepted as an infectious model of RSV; we have added a few of references to document this ([26-28]) (Section 3.1 line 124; Section 3.3 line 279). The Reviewer appears not to be aware that our recent eLife paper (Ref 19) reported strong effects on mitochondrial respiration, mtROS production, and protection against these effects by the mtROS scavenger MitoQ in A549 and primary human bronchial epithelial cells, essentially in identical fashion to the results here for HEK293Ts; we also showed that MitoQ protects against infectious virus production in A549 and primary human bronchial epithelial cells as well as in an animal model. We have now made this clear in the manuscript, where appropriate (eg. Section 3.3 lines 279-282), as well as in the Discussion section (lines 353-368). To satisfy the Reviewer, however, we have included a new figure (Figure 6), highlighting MitoQ’s strong protective ability against RSV infection in A549 cells (Section 3.3 lines 282-289); the results are essentially identical to those for HEK293T cells (Figure 5), underlining the robustness of the HEK293T system.
Description of statistical analysis in the methods is incomplete (which type of ANOVA was used?) and it’s
lacking in the legend of Fig. 1.
We thank the Reviewer for pointing out our omission; we have added the information to the manuscript (Section 2.5 lines 115/6; Figure 1 lines 155/6).
Discussion is very short, 20 lines in length and does not really try to connect their observation with the know role of mitochondria in virus-induced numerous innate immune responses, as many steps of virus-induced signaling cascades occur on mitochondria or require mitochondrial components, or the role of mitochondria in other cellular process like apoptosis, all of which are known to modulate virus replication. The metabolic state of the mitochondria and its bioenergetics is also linked to virus replication in multiple ways.
AS requested, we have expanded the discussion on the physiological relevance of host cell mitochondria bioenergetics, apoptosis, and antiviral signaling to RSV replication/production (Section 4 lines 322-336), and protection against RSV infection using mitochondrial ROS scavenger MitoQ, including referring to our previous results using a mouse model (Section 4 lines 353-368).
We thank the Reviewers again for their important contributions to making our manuscript a more rigorous and clearer evocation of work that is of interest to the broad readership of Cells.
We thank you for your assistance, and in anticipation of receiving acknowledgment of receipt of the manuscript.
Yours sincerely,
Prof. David A. Jans
PS. In addition to the revised manuscript (“Main Article File”) we have submitted a “Related Manuscript File” that highlights in red all of our changes to the original manuscript listed above. We have also submitted a Supplementary Material file which is all new data.
PPS. We have now put all of our references into the MDPI Citations Style Guide, as requested.
Round 2
Reviewer 2 Report
The authors have responded sufficiently to the comments